# Unraveling the Multifaceted Role of the Golgi Apparatus: Insights into Neuronal Plasticity, Development, Neurogenesis, Alzheimer’s Disease, and SARS-CoV-2 Interactions

**DOI:** 10.3390/brainsci13101363

**Published:** 2023-09-23

**Authors:** Corneliu Toader, Lucian Eva, Razvan-Adrian Covache-Busuioc, Horia Petre Costin, Luca-Andrei Glavan, Antonio Daniel Corlatescu, Alexandru Vlad Ciurea

**Affiliations:** 1Department of Neurosurgery, “Carol Davila” University of Medicine and Pharmacy, 020021 Bucharest, Romania; corneliu.toader@umfcd.ro (C.T.); horia-petre.costin0720@stud.umfcd.ro (H.P.C.); luca-andrei.glavan0720@stud.umfcd.ro (L.-A.G.); antonio.corlatescu0920@stud.umfcd.ro (A.D.C.); prof.avciurea@gmail.com (A.V.C.); 2Department of Vascular Neurosurgery, National Institute of Neurology and Neurovascular Diseases, 077160 Bucharest, Romania; 3Faculty of Medicine, “Dunarea de Jos” University of Galati, 800201 Galați, Romania; 4Emergency Clinical Hospital “Prof. dr. N. Oblu”, 700309 Iasi, Romania; 5Neurosurgery Department, Sanador Clinical Hospital, 010991 Bucharest, Romania

**Keywords:** Golgi apparatus, neuronal plasticity, neuronal development, neurogenesis, Alzheimer’s disease, SARS-CoV-2

## Abstract

This article critically evaluates the multifunctional role of the Golgi apparatus within neurological paradigms. We succinctly highlight its influence on neuronal plasticity, development, and the vital trafficking and sorting mechanisms for proteins and lipids. The discourse further navigates to its regulatory prominence in neurogenesis and its implications in Alzheimer’s Disease pathogenesis. The emerging nexus between the Golgi apparatus and SARS-CoV-2 underscores its potential in viral replication processes. This consolidation accentuates the Golgi apparatus’s centrality in neurobiology and its intersections with both neurodegenerative and viral pathologies. In essence, understanding the Golgi’s multifaceted functions harbors profound implications for future therapeutic innovations in neurological and viral afflictions.

## 1. Introduction

The complexity of the nervous system’s architecture is underpinned by processes of neuronal maturation and synaptic adaptability. Central to these intricate mechanisms is an oft-overlooked cellular entity in neurobiological discussions: the Golgi apparatus. Functioning as the cellular “sorting hub”, the Golgi apparatus is indispensable in protein categorization and adjustments crucial to the dynamic equilibrium of neuronal structure, function, and synaptic adaptability [1]. Contemporary research has illuminated potential associations between the Golgi apparatus and the pathogenesis of Alzheimer’s, particularly emphasizing the role of the Golgi matrix protein GM130 (a morphological determinant protein situated on the cis-face of the Golgi apparatus) [2,3,4]. Novel insights are surfacing, linking this protein to its Golgi-based counterpart and the pathological shifts characteristic of Alzheimer’s, thus ushering in innovative investigative trajectories in our pursuit of understanding this devastating neurodegenerative disorder [4].

The Golgi apparatus is not only pivotal in protein categorization and alterations but also stands out as a central orchestrator of neuronal maturation. The nuanced orchestration of axonal and dendritic differentiation is facilitated by this organelle, paving the way for growth and distinctiveness within neural circuits. Moreover, its influence is unmistakably evident in synaptic plasticity, fundamental processes governing learning and memory within the nervous system [5].

Overall, this paper aims to portray the importance of the Golgi apparatus in key aspects of neurobiology, such as neurogenesis, neural development, and neural plasticity, while also elucidating the pathophysiology of the Golgi apparatus in SARS-CoV-2 infection and Alzheimer’s disease.

## 2. Golgi Apparatus in Axonal Development

Cerebral cells, comprising neurons and glia, are characterized by intricate architectures tailored to execute designated roles. Neurons establish synapses to facilitate electrical communication, oligodendrocytes envelop axons with myelin sheaths for insulating properties, and astrocytes bridge vascular networks [6,7]. These specialized morphologies necessitate distinct routes for protein trafficking, prominently featuring the Golgi apparatus outposts (GOPs) (Figure 1). In specialized cellular contexts, Golgi outposts assume critical functions essential for the sculpting of distinct cellular morphologies and architectures. Notably, within neurons, these organelles are pivotal for the formation of dendritic branches [8]. Concurrently, in muscle cells, they are implicated in the genesis of grid-like microtubule lattices, and in oligodendrocytes, they facilitate the formation of microtubules that encircle the myelin sheath in a spiral configuration [9]. Intriguingly, notwithstanding their crucial roles, to date, Golgi outposts have been exclusively identified in vivo and within primary cultured cells, evading detection in immortalized cell lines [10].

In summary, the secretory pathway initiates in the endoplasmic reticulum (ER), where nascent proteins are synthesized. Subsequently, these proteins traverse through the Golgi apparatus (GA) for further maturation, processing, and sorting. They are then dispatched to subsequent post-Golgi/trans-Golgi network (TGN) compartments, ultimately reaching their designated locations such as endosomes, lysosomes, and the plasma membrane [11]. Transmembrane proteins synthesized on the RER are incorporated into the ER membrane subsequently exiting through specialized sites termed ERES (ER exit site) with the aid of COPII vesicles [12]. Following the detachment of COPII proteins [13], these vesicles employ SNAREs (Soluble NSF Attachment Protein Receptors) to amalgamate with tubular assemblies termed ER-Golgi intermediate compartments (ERGICs). Additionally, certain proteins can retrogress from the ER via COPI vesicles initiated at the Golgi apparatus. Within the ERGIC, proteins undergo processing and subsequently navigate to the Golgi apparatus for further modifications, encompassing glycosylation and proteolysis. This journey encompasses the cis-Golgi network (CGN), medial Golgi cisternae, and the trans-Golgi network (TGN) [14].

Upon reaching the TGN, proteins adopt diverse processing trajectories contingent upon their intended function and localization. Vesicles emanating from the Golgi, laden with secreted or transmembrane proteins, can bifurcate and merge with plasma membranes, undergoing subsequent processing. This can culminate in the extracellular secretion of proteins or their integration into plasma membranes. Additionally, vesicles laden with secretory proteins can be conserved proximal to plasma membranes or endosomes, awaiting release upon specific stimuli. Moreover, proteins destined for endosomes and lysosomes undergo modification via an oligosaccharide marker termed mannose 6-phosphate, and subsequently depart the TGN enclosed within clathrin-coated vesicles [15]. Consequently, the selection of secretory routes is determined by the intended role, ultimate destination, and inherent nature of the respective cargo proteins [16].

## 3. Golgi Apparatus Involvement in Dendritic Formation

Neuronal maturation is characterized by the augmentation of the plasma membrane during processes such as polarization and outgrowth. The localization of membrane constituents is contingent upon the secretory pathway, a sequence encompassing a myriad of organelles such as the endoplasmic reticulum (ER), ER-Golgi intermediate compartment, cis-Golgi, medial Golgi, and trans-Golgi, among others [17]. These specialized organelles orchestrate the synthesis and delivery of novel membrane lipids and proteins; however, the understanding of their spatial arrangement, functionality, and regulatory mechanisms within neurons remains nascent.

In non-neuronal cells, the architecture of the secretory pathway organelles is notably conserved. The ER permeates the cell and includes distinct locales termed “ER exit sites”. Here, the coat protein complex II-coated vesicles, laden with newly synthesized cargo, commence their journey to the Golgi complex for subsequent protein processing and sorting. Ultimately, post-Golgi carriers facilitate the conveyance to the plasma membrane [18,19].

Contemporary research underscores that dendrites, the specialized neuronal protrusions, possess inherent secretory faculties. For instance, vesicles emanating from the trans-Golgi network partake in calcium-mediated exocytosis within dendrites [20]. Notably, glycine and glutamate receptors are rapidly discernible on dendritic surfaces, potentially originating from dendritic compartments housing secretory proteins. Dendrites also exhibit the ability to integrate sugars and translate membrane protein-encoding mRNA, hinting at their Golgi-mimetic functions [21]. However, it remains an enigma whether dendritic trafficking encompasses the entire secretory continuum, from ER to Golgi, or is exclusive to the latter stages involving post-Golgi vesicles.

Dendritic structures house distinct Golgi units adept at secretory cargo transport [21]. Quantitative analyses reveal that within the hippocampal neuronal cohort studied, 70% exhibited dendritic Golgi apparatus. An intriguing observation highlighted that most neurons had Golgi units localized to a singular dendrite, while a minor proportion exhibited multiple dendrites with Golgi presence. Overall, 51% of hippocampal neurons possessed Golgi in at least one dendrite, 19.5% in two, and 29.5% were devoid of dendritic Golgi [22]. Interestingly, the presence or absence of Golgi in specific dendrites was not contingent upon the diameter of their proximal extensions.

Contrasting starkly with other membrane-bound organelles ubiquitously found within dendrites, such as ER exit sites, endosomes, and mitochondria, Golgi bodies in dendrites presented a distinct pattern. Utilizing GM130 as a marker, Golgi stacks were conspicuously identified in proximal regions of pyramidal neurons’ apical dendrites but were seldom evident on basolateral extensions [23].

A limited subset of neuronal dendrites houses Golgi outposts, typically restricted to one per neuron. These outposts have garnered significant scholarly attention, especially given their potential role as non-centrosomal microtubule organizing centers (MTOCs). Outposts are capable of synthesizing microtubules at considerable distances from their nuclei, facilitating the establishment of microtubule networks. Such configurations are evident in cells with distinct structural attributes, as seen in *Drosophila* neurons, murine muscle cells, and rodent oligodendrocytes. Furthermore, Golgi outposts have been implicated in various pathologies, including muscular dystrophy and Parkinson’s disease. Their distribution is especially pronounced in pyramidal neurons, where they predominantly localize to the apical dendrite [24].

The elaboration and extension of dendrites and axons potentially necessitate coordinated modifications in cytoskeletal organization and membrane transport mechanisms. Specifically, molecules such as RhoA [25] and MAP2 [26] appear to be predisposed to augment dendritic growth. Concurrently, a commensurate adaptation in membrane provisioning might be requisite to facilitate their distinctive morphologies.

Intriguingly, neurons also harbor unique secretory conduits in dendrites, which encompass both ER and Golgi outposts [16]. Yet, a conspicuous absence of Golgi outposts in axons prompts inquiries regarding the potential role of the polarized distribution of Golgi proteins in delineating the differential maturation trajectories of dendrites and axons. Contemporary research intimates a proactive involvement of secretory trafficking in the genesis of specialized dendritic compartments. Nevertheless, the extent to which these secretory pathways modulate the differential elongation patterns of dendrites and axons remains enigmatic (as illustrated in Figure 2). Furthermore, the implications of dendritic Golgi outposts in dendritic elaboration, and their putative role in sculpting the distinct morphological attributes of dendrites and axons, await elucidation [27].

Imaging analyses on cultured hippocampal neurons have elucidated the pivotal role of dendritic Golgi outposts (GOPs) in trafficking subsequent to the endoplasmic reticulum, notably via the observation of GFP-VSV-G ts045 transport [27,28]. Intriguingly, NMDA receptors seem to follow a distinct sorting pathway from AMPA receptors at the ER, predominantly associating with dendritic GOPs instead of transiting through somatic Golgi apparatuses (GAs). This intimates that mini-GAs within GOPs might serve as strategic platforms to locally dispatch synaptic receptors, potentially facilitating synaptic plasticity [29,30]. GOPs are instrumental in influencing dendritic expansion and branching dynamics [29,30], likely by orchestrating the allocation of cargo to disparate dendritic branches [28]. Investigations on *Drosophila* sensory neurons have illuminated that perturbations in the Lava-lamp adaptor, or its mutations, culminate in diminished GOPs and a consequent reduction in dendritic branches [31]. For *Drosophila* dendritic arborization neurons, it has been posited that GOPs orchestrate dendritic morphology by acting as epicenters for microtubule nucleation [30]. However, contemporary findings have emphasized that γ-tubulin actually governs dendritic microtubule nucleation, independent of GOPs, across diverse *Drosophila* neuronal classes [32].

Pertaining to GOP genesis, a plausible mechanism might encompass localized de novo synthesis from the ER, reminiscent of the Golgi reconstitution observed at cellular extremities in non-neuronal entities [33]. Alternatively, GOPs could emerge post somatic Golgi fragmentation due to heightened neuronal activity, with remnants of the degraded Golgi potentially serving as blueprints for the subsequent assembly of satellite Golgi arrays [34]. Formation of GOPs might also ensue from the primary somatic GA, prompting inquiries into the participation of Golgi fission machinery components such as LIMK1 [35] and Protein Kinase D1 (PKD1) [36], and the potential involvement of their proximal regulators and distal effectors. Notably, evidence underscores the potential influence of a RhoA-Rho kinase (Rock) signaling cascade in the orchestration of polarized GOPs during neuronal morphogenesis.

The Golgi apparatus inherently harbors the capability to act as a reservoir of membrane constituents vital for localized cellular expansion [27]. Dendrites replete with Golgi elements consistently manifest greater length and intricate architecture as compared to their Golgi-devoid counterparts. Empirical assessments have ascertained that an overwhelming 86% of neurons under examination showcased Golgi elements within their most elongated dendrite [22].

In mature neurons, a hallmark characteristic resides in their intricately branched dendritic arbors, which conspicuously surpass the complexity of dendrites devoid of Golgi outposts.

Collectively, these revelations accentuate that a non-uniform Golgi distribution resonates with dendritic expansion patterns. The enhanced length and complexity of Golgi-endowed dendrites underscore the indispensable role of this organelle in sculpting dendritic architecture.

## 4. BARS Regulation of Golgi Trafficking

The CtBP (C-terminal-binding protein) family, comprising CtBP1 and CtBP2, is pivotal in a gamut of biological undertakings such as development, differentiation, oncogenesis, and apoptosis, chiefly functioning as transcriptional co-repressors within the nuclear compartment [37]. BARS (Brefeldin A ADP-Ribosylated Substrate) emerges as a salient mediator in membrane fission activities including macropinocytosis, fluid-phase endocytosis, COPI-coated vesicle genesis, and an array of post-Golgi carrier and ribbon events during mitosis. The intricacies of BARS in fission are well-chronicled in cellular studies, where it is posited as a key intermediary in basolateral post-Golgi carrier formation. Specifically, MDCK cells, manifesting polarized epithelial attributes, display hindered Vesicular Stomatitis Virus G (VSVG) transport upon BARS inhibition, attributed to tubular carriers laden with this payload remaining adhered to the Trans Golgi Network (TGN). This fission process orchestrated by BARS at the TGN entails a multifaceted ensemble of proteins [38,39].

Through in situ hybridization evaluations, it is apparent that CtBP family constituents are profusely expressed across the neural framework. Notably, CtBP1 and CtBP2 showcase divergent distribution spectra within mature cerebral structures, distinguished by expression magnitudes, territorial expression blueprints, subcellular targeting proficiencies, and intracellular positioning paradigms. Mouse-based investigations with deletions in one or both CtBP genes intimate the existence of both common and unique functionalities for these proteins [40,41]. Of particular significance is the discernment that concurrent excision of CtBP1 and CtBP2 in murine models precipitates retarded development of forebrain and midbrain structures, a sequela of modulated transcription factor activities in their absence [42,43].

Diminished BARS expression is correlated with marked curtailment in hippocampal neuronal growth, migration, and the multipolar-to-bipolar transitional dynamics in cortical neurons. Notably, these effects can be mitigated by co-expression of fission-inefficient mutants exhibiting reduced nuclear positioning or fission vigor. Contrarily, accentuated growth is observed upon co-expression of the said mutants, an effect which can be counteracted by the concurrent expression of aforementioned mutants [39].

The meticulous orchestration of membrane constituents along the secretory trajectory is cardinal for neuronal tasks such as augmentation, polarity inception, sustenance, synaptic plasticity, and cellular migration [44,45]. This conduit is initiated at the rough endoplasmic reticulum, subsequently traversing the Golgi machinery where membrane proteins designated for axons and dendrites are segregated into specific carriers for extracellular dispatch [46]. Upon departure from the trans-Golgi matrix, they are relayed to plasma membranes via molecular propellants, exemplified by kinesin superfamily entities and myosins. Despite this knowledge, the specialized apparatus governing their TGN exit remains inadequately delineated. Nevertheless, it is established that BARS, an integral affiliate of the CtBP assemblage, is indispensable during neuronal ontogeny [47].

Both LIMK1 and PKD1 are quintessential components of the TGN fission apparatus in polarized epithelial structures, overseeing membrane protein transference either to apical or basolateral facets [48,49]. Concurrently, these proteins, evident within neurons situated at GAs, partake in dynamin-facilitated fission of Golgi conduits, thereby engendering Golgi outposts. Ensuing research ought to delve into the mechanisms by which BARS engenders dendritic carrier fissions at TGNs and probe the putative role of BARS in the fission of Golgi conduits during the generation of Golgi outposts [50].

## 5. Golgi Apparatus and Synaptic Plasticity

Rare diseases, particularly genetic anomalies, exert a pronounced effect on the nervous system, culminating in manifestations such as neurodegeneration and behavioral perturbations [51]. These infrequent neurological conditions furnish a platform to elucidate hitherto uncharted cellular dynamics integral to neuronal performance. This is especially pertinent in disorders associated with ATP7A/ATP7B genes and COG complex subunit genes. The proteins expressed from these genes predominantly localize within the Golgi apparatus under standard conditions [52]. Over the course of evolution, organisms have intricately modulated copper metabolism and transport. Central to this regulatory framework are ATP7A and ATP7B proteins. These proteins, classified under P-type Cu-transporting ATPases, harness ATP hydrolysis to facilitate the transmembrane movement of copper ions. Essentially, these copper pumps either extrude surplus copper from cells or allocate it to copper-reliant enzymes. The physiological roles of Cu-transporting ATPases are multifaceted, spanning dietary copper excretion through bile, placental transport, and lactational secretion to modulating resistance against select anti-cancer therapeutics [53,54,55,56].

Mutational alterations in the ATP7A gene underpin Menkes disease, typified by systemic copper deficiency stemming from hindered intestinal copper assimilation [57]. The clinical picture of Menkes disease predominantly emerges in childhood, exhibiting a spectrum of systemic and neurologically linked symptoms. The latter include intellectual impairments and gray matter neurodegeneration [58]. Such clinical manifestations may be ascribed to perturbations in copper-reliant enzymes that traverse the Golgi apparatus or reside within mitochondria [59]. Paradoxically, cell-specific aberrations in ATP7A precipitate intracellular copper accumulation, a scenario arising despite the overarching copper deficiency evident in Menkes patients [60]. In a contrasting paradigm, mutations in the ATP7B gene underlie Wilson’s disease. This disorder is marked by hepatic impediments to copper excretion, ensuing in systemic copper excess, hepatotoxicity, psychiatric manifestations, and neurodegeneration, particularly in the lenticular domain [59].

Differential expressions of ATP7 and COG complex subunits can significantly modulate synaptic morphology, mitochondrial constituents, and neurotransmission in response to stimulatory or plasticity-inducing cues. A deeper exploration is imperative to elucidate these interconnected dynamics. Mitochondria, pivotal cellular organelles, mediate synaptic operations through diverse modalities. They serve as calcium reservoirs [61], facilitate ATP synthesis [62], generate Krebs cycle intermediates influencing neurotransmission [63], and engage in glutamate metabolism [64]. Additionally, mitochondria confer metabolic adaptability by diversifying carbon inputs into the Krebs cycle, including sources such as pyruvate, glutamate, or fatty acids [64].

The mitochondria’s competence in modulating synaptic calcium concentrations might shed light on certain neurotransmission phenotypes, particularly in contexts of ATP7 overexpression or disrupted copper efflux. Brief episodes of high-frequency neuronal excitation can engender synaptic plasticity via facilitation, augmentation, and potentiation, cumulatively enhancing transmission efficiency, termed synaptic enhancement [65]. Facilitation hinges on the preservation of optimal calcium concentrations post-influx, while concurrently augmenting it with basal calcium during neuronal activity and inducing exocytosis of readily releasable vesicles in response to neural excitation [66,67,68]. In contrast, potentiation seeks a balance between exocytosis and endocytosis [69].

Post-tetanic facilitation and augmentation typically involve a diminution in residual calcium [65]. Basal calcium concentrations appear cardinal for post-tetanic potentiation (PTP), with activity-mediated calcium elevations instigating phosphorylation-oriented signaling persisting beyond its ephemeral phase [69]. Given the involvement of mitochondria during PTP evolution, their potential role here merits consideration [70,71].

Mitochondria’s indispensable role in modulating calcium concentrations at neuronal terminals is paramount for synaptic plasticity and the inception of short-term memories. They act as calcium buffers postconditioning [72]. However, in synapses manifesting ATP7 overexpression or compromised copper efflux pathways, any enhancements post-sustained tetanic or post-tetanic stimuli are negated. It is salient that upon diminishing COG complex expression, both mitochondrial content and neurotransmission phenotypes in ATP7 hyperexpressing terminals are concurrently ameliorated, substantiating the hypothesis of mitochondria-mediated mechanisms dictating these phenotypes [73,74].

## 6. LARGE Gene Interactions with Golgi Apparatus—Consequences and Implications

The LARGE gene, prominently expressed in the hippocampus compared to other tissues, plays a pivotal role in cognitive functions [61]. Mutations within this gene give rise to human congenital muscular dystrophy type 1D, characterized by severe cognitive impairments, atypical electroretinogram results, and nuanced structural cerebral anomalies [62,63]. LARGEmyd mice, harboring truncation mutations of the LARGE gene, exhibit neurological manifestations akin to humans bearing such mutations, including sensorineural hearing loss, retinal transmission deficits, neurodevelopmental irregularities, and attenuated long-term potentiation [LTP] [64,65]. Recent research underscores the potential of aberrant synaptic operations as the underlying cause of such cognitive impediments [64].

Intellectual disability, historically termed mental retardation, is delineated by pronounced cognitive (IQ ≤ 70) and adaptive behavior deficits, manifested as restricted conceptual, social, or practical skills prior to 18 years of age [75]. This disability might also correlate with other cognitive deteriorations across one’s lifetime, including dementia of a neurodegenerative origin. Numerous X-linked cerebral disorders related to intellectual disability have proteins, instrumental in synaptic signaling pathways, as plausible etiological agents [76,77,78]. Fragile X syndrome, a predominant genetically inherited intellectual disability, is directly associated with AMPA-R dysregulation. Proteins PAK3 and OPHN1, implicated in intellectual disability, oversee synaptic AMPA-R expression and stability [74]. Mutations in its GluA3 subunit have been identified in individuals with X-linked intellectual disability [79]. However, the specifics of these perturbations in AMPA-R dynamics and their contribution to cognitive dysfunction remain largely enigmatic.

The LARGE gene is instrumental in regulating AMPA-R—a synaptic receptor crucial for synaptic plasticity and long-term potentiation, vital for memory consolidation and learning. Mice devoid of this protein manifest compromised LTP, potentially correlated with neurodevelopmental aberrations such as anomalous neuronal migratory patterns [80].

Experiments aimed at excising dystroglycan from murine brains yield outcomes reminiscent of congenital muscular dystrophies, such as lissencephaly. This implies a central role of dystroglycan in the central nervous system, suggesting that the anomalies observed in LARGEmyd mice might echo mechanisms operational in humans with similar dystrophies [81].

LARGEmyd mice present with irregular stratification within their cerebral and cerebellar structures. Atypical neurons are evident across both regions. Additionally, traces of neurons in the external granular layer, presumably attributable to migratory defects, are observed along with congregations of inappropriately located neurons within the white matter or beneath the pial surface. Such abnormalities mirror those seen in specific muscular dystrophies, including Fukuyama congenital muscular dystrophy (FCMD) and Muscle–Eye–Brain disease (MEB) [63,82,83].

Microscopic evaluations have identified dystroglycan in the hippocampus, particularly within postsynaptic assemblies crafted by mossy fibers on pyramidal neurons [84]. Prior investigations revealed that dystroglycan suppression, achieved using GFAP-Cre, results in diminished LTP, proposing its role in synaptic functionality [85]. Synaptic adaptability alterations might arise due to diminished dystroglycan in either neurons or glial cells—fundamental to the orchestration of synaptic operations [86]. The potential contribution of developmental cerebral anomalies in influencing the electrophysiological attributes of the GFAP-Cre/DG-null brain cannot be dismissed [64].

Moreover, research has pinpointed the LARGE protein’s role in thwarting AMPA-R localization by impeding its transit from the Golgi apparatus to the cellular exterior. A deficiency in LARGE culminates in an AMPA-R overabundance at synapses, hindering hippocampal LTP. Experiments utilizing animal models exposed to both knockout and knockdown vectors bearing small hairpin RNA target LARGE unveil defects in associative fear memory formation. This highlights LARGE’s significance in memory creation through its governance of AMPA-R movement from the Golgi to the cellular facade [80]. Collectively, these insights emphasize its cardinal role in memory genesis by modulating synaptic AMPA-R localization within the hippocampus [87].

## 7. Golgi Matrix Protein 130 (GM130)

### 7.1. Role of GM130 in the Golgi Apparatus

The Golgi Matrix Protein 130 (GM130) emerged as the inaugural matrix protein deemed indispensable for preserving the Golgi apparatus’s morphology, identified in a 1995 screen for proteins affiliated with the Golgi apparatus (GA) [88]. The GOLGA2 gene is responsible for the transcription of GM130, a Golgin protein that has been the focus of extensive investigations. Acting as a Golgi architectural protein, GM130 adheres firmly to Golgi membranes, safeguarding their sophisticated configuration. Additionally, GM130 is integral to several cellular endeavors, encompassing: vesicular fusion between the Golgi and ER [89]; mitotic spindle organization and cellular bifurcation; microtubule nucleation at the Golgi; and compartmental layout in dendritic Golgi outposts [90].

### 7.2. Pathological Dimensions of GM130 

Mutations culminating in GM130 inactivation or diminished expression have been identified in patients grappling with diverse ailments, including colorectal and breast cancers. Intriguingly, an elevated GM130 expression in gastric cancer correlates with reduced patient survival [91]. Diacylglycerol acyltransferase I (DGAT1) may serve as a therapeutic modality against prostate cancer by adjusting microtubule-organizing centers and GM130 concentrations, thereby undermining microtubule structural integrity [92,93]. The presence of Mannose N-glycans and the Golgi residency of α-mannosidase 1A could serve as prognostic markers for aggressive prostate carcinoma cells [94]. Interference in GM130 and GRASP65 binding precipitates its degradation, triggering Golgi fragmentation and acute pancreatitis in murine models [95]. Cells bereft of NAGLU exhibit heightened GM130 expression, leading to the elongation and expansion of the Golgi and anomalous lysosomal accrual; restoring GM130 levels can ameliorate these pathological hallmarks [96]. The mammalian GA is instrumental in modulating surfactant protein secretion by pulmonary epithelial cells [97].

### 7.3. Neurological Implications of GM130 Dysfunction

Furthermore, GM130 ablation in murine nervous systems can precipitate the gradual demise of Purkinje neurons in the cerebellum, resulting in discernible motor dysfunctions, imbalanced postures, and severe tail-suspension rotational behaviors, mirroring cerebellar ataxia. In advanced stages, paralytic manifestations with degenerative traits can be evident. Zebrafish bearing mutations that deactivate GM130 demonstrate pronounced skeletal muscle anomalies and progressive microcephaly (MCPH) [2]. Individuals harboring homozygous variants of such mutations manifest analogous clinical presentations, including myofibrillar degeneration, hypotonia, growth impediments, and degenerative features [98].

GM130 exerts a fundamental influence on nervous system ontogeny, orchestrating myriad biological operations vital for its optimal functionality. Among these are: preservation of the Golgi apparatus’s structural integrity, efficient intracellular protein and lipid trafficking, mitotic activities such as mitotic segregation, and regulatory mechanisms overseeing cellular migration/polarization, along with proficient glycosylation processes [99].

The Golgi apparatus, a cornerstone of the endomembrane network, exhibits structural deviations linked to assorted neurodegenerative conditions. As an intrinsic constituent of the GA matrix, GM130 is paramount in conserving its ribbon-esque architecture [100]. An aberrant GA morphology frequently coincides with a diminished matrix protein expression. The assembly of the Golgi ribbon necessitates seamless integration of ER-to-Golgi carriers (EGCs or ERGICS) into GA layers, contingent on the uninterrupted shuttling of GM130 between the cis-Golgi and EGCs. In the absence of GM130, this process falters, leading to vesicular membrane accumulation, curtailment of smooth ER vesicles, and Golgi ribbon disintegration [101].

## 8. Alzheimer’s Disease and GM130

### 8.1. Neuronal Development and the Role of GM130 in Neurodegenerative Diseases

During embryogenesis of neurons, the Golgi apparatus emerges as a pivotal non-centrosome-associated microtubule organizing center. Cells forming dendrites leverage these structures for direct cargo transport to nascent dendritic plasma membranes and localized microtubule nucleation to support dendrite elongation [23,102]. Neurodegenerative conditions, including Alzheimer’s Disease (AD), Parkinson’s Disease (PD), Amyotrophic Lateral Sclerosis (ALS), and Spinocerebellar Ataxia Type 2 (SCA2), are characterized by a disrupted Golgi architecture. GM130 ensures the integrity of the Golgi’s structure, anchoring transport vesicles crucial for streamlined endoplasmic reticulum (ER)-to-Golgi transport [101]. Furthermore, GM130 is indispensable for precise Golgi positioning, cytoskeletal modulation, and establishment of neuronal Golgi satellites [23]. Aberrations in GM130 functionality, manifested as vesicular accumulation and Golgi disarray, correlate with neuronal impairment and cell apoptosis [103]. Investigations utilizing human induced pluripotent stem cells and neurons with compromised GM130 functionality elucidate adverse repercussions on cell polarity, motility, migration, neurogenesis, and neuritogenesis [104]. Given its multifaceted physiological implications, GM130 is instrumental in a plethora of neuropathological conditions.

### 8.2. Alzheimer’s Disease: Pathogenesis and Golgi Disruption

Alzheimer’s Disease is characterized as a progressive, age-associated cognitive deterioration paired with distinct neuropathological signatures. Initial manifestations encompass diminished memory consolidation capabilities, which evolve into broader cognitive and behavioral anomalies. Contemporary therapeutic strategies for AD encompass acetylcholinesterase inhibitors for cognitive augmentation and nonsteroidal anti-inflammatory agents to potentially decelerate disease progression and ameliorate cognitive deficits [105].

Central to AD pathogenesis is the aggregation of amyloid-β (Aβ) peptides, derivatives of amyloid precursor protein (APP) cleavage, implicated in neurodegenerative pathways in AD patients [106,107]. Aβ peptides, functioning as cell surface ligands, are pivotal for neurite extension, cellular adhesion, and synaptogenesis, among other cellular processes [108]. Fluorescent microscopic analyses of hippocampal specimens from transgenic mice models of AD, possessing the APP Sweden and PS1 deletion mutations, highlighted disrupted Golgi morphologies in contrast to the intact perinuclear formations in control mice [109,110,111,112]. Additionally, early AD progression is marked by neuronal Golgi fragmentation and spatial redistribution. Ultrastructural assessments delineate perturbed Golgi lamellae with diminished diameters and proximal vesicular congregates [113]. Such Golgi disorganization could compromise protein trafficking dynamics across Golgi membranes, adversely affecting APP processing and elevating Aβ synthesis [114]. Herein, GM130’s core functionality in preserving Golgi architecture becomes evident.

Cdk5 appears to be instrumental in AD-associated Golgi disintegration. Evidenced by Cdk5 phosphorylation loci on GM130, unchecked Cdk5 activity might instigate Golgi fragmentation, acting as a substrate for Cdc2. Analogous to the early prophase, where Cdc2-mediated GM130 phosphorylation persists through metaphase to anaphase culminating in fragmentation, with subsequent dephosphorylation in telophase promoting Golgi reconstitution, Cdk5 might exert a similar modulatory influence [115].

## 9. Interplay between the Golgi Apparatus and SARS-CoV-2: Possible Associations with Alzheimer’s Disease

The enveloped SARS-CoV-2 virus exploits the host cell’s secretory pathway for its lifecycle processes, including replication, assembly, and egress. Within the host cellular environment, the virus harnesses three non-structural proteins, namely, Nsp3, 4, and 6, to reconfigure the endoplasmic reticulum, resulting in the creation of double-membrane vesicles instrumental for viral RNA replication. The ER-Golgi intermediate compartment (ERGIC) and the Golgi apparatus serve as the milieu for virion assembly, where processes such as spike protein glycosylation and furin cleavage transpire. Efficient functioning of this assembly hub is contingent on a protein stack formed by GRASP55 and GRASP65. Deficiency in these GRASP proteins culminates in Golgi fragmentation (GF), enhancing trafficking, albeit with ramifications including altered lysosomal enzyme sorting, protein glycosylation modifications, and disruptions to cellular functions such as adhesion and proliferation [116,117,118].

In the context of SARS-CoV-2 infections, GRASP55, integral to various stress responses, undergoes down-regulation, whereas GRASP65 remains largely stable [119]. Research underscores GRASP55’s salience in viral infections, potentially attributable to its role in Golgi architecture. Corroborating this, diminished GRASP55 expression impedes the translocation of the spike protein to the cell’s exterior, implying that virus-induced down-regulation of GRASP55 expedites viral trafficking [120]. Such observations align with prior work highlighting increased protein trafficking in the wake of GRASP55 deficiency [121]. Furthermore, GRASP55 expression diminishes ACE2 surface presence, insinuating its potential impact on the SARS-CoV-2 entry pathway.

Electron microscopy has elucidated marked disparities in Golgi configurations between infected and non-infected cells. The Golgi apparatus in non-infected cells exhibits a pronounced density proximal to the nucleus with extended cisternae. In contrast, SARS-CoV-2-afflicted cells display substantial Golgi disarray, with predominant vesiculation of Golgi membranes. Intriguingly, copious viral particles inhabit the expansive Golgi lumen, suggesting the virus’s disruptive transit through the Golgi [122]. Moreover, in a neuropathological examination of 20 COVID-19 cases, six (three biopsies and three autopsies) exhibited white matter abnormalities on MRI, presenting microhemorrhages suggestive of small artery diseases. These cases displayed COVID-19-associated cerebral microangiopathy (CCM) with distinct perivascular changes and evidence of blood–brain barrier compromise. Despite the absence of fibrinoid necrosis and viral RNA in the brain, the SARS-CoV-2 spike protein was identified in the brain endothelial cells’ Golgi apparatus, associating with the furin protease. Cultured endothelial cells resisted SARS-CoV-2 replication, and the protein’s distribution differed from that in pneumocytes [123].

Alzheimer’s disease (AD) presents a parallel, with its brain specimens often exhibiting Golgi fragmentation, attributed to heightened neuronal activity. Golgi anomalies are implicated in several AD-related pathologies, encompassing aberrant protein sorting and glycosylation, hindered lysosomal/autophagosomal degradation, and an escalation in Aβ peptide production. Notably, Aβ oligomer accumulation triggers GF through cyclin-dependent protein kinase 5 (CDK5) activation, which subsequently phosphorylates GRASP65, elevating Golgi growth factor. This series of events amplifies APP trafficking and, in turn, Aβ peptide production [124].

GRASP65’s role is pivotal in mitigating amyloid-beta generation by promoting nonamyloidogenic a-cleavage of the amyloid precursor protein. This unveils an intricate interplay between augmented amyloidogenic cleavage of APP and tau phosphorylation, with CDK5 being the primary kinase regulating tau phosphorylation and influencing Aβ formation [125].

Conversely, GRASP55 modulates the secretion and aggregation of neurotoxic proteins via autophagy. Ordinarily, GRASP55 undergoes O-linked N-acetylglucosamine (O-GlcNAc) modification, impacting nucleocytoplasmic proteins. However, under glucose-deprivation, GRASP55’s O-GlcNAcylation wanes, prompting its relocation to function as a membrane tether for autophagosome–lysosome fusion, thereby augmenting unconventional secretion of neurotoxic entities, such as tau [106,126,127,128].

Appreciating the ramifications of SARS-CoV-2 infection on neural cells at the molecular and cellular echelons remains paramount. With Golgi irregularities providing an investigative fulcrum, probing their association with Alzheimer’s pathophysiology and clinical dementia mandates delving into the molecular underpinnings. Such endeavors can elucidate mechanisms wherein Aβ fibrils interfere with protein trafficking across neural cell types, linking observed aberrations in glycosylation and lysosomal activity to these disruptions (Figure 3).

## 10. Conclusions

The Golgi apparatus, frequently marginalized in neurobiological discourse, has recently emerged as pivotal to a plethora of processes within the nervous system. This intricate organelle is integral to multifaceted biological phenomena encompassing neurogenesis, Alzheimer’s Disease, and SARS-CoV-2 infections.

Its cardinal function in axonal and dendritic dynamics, as well as synaptic plasticity, underscores its indispensability during neuronal maturation. Such involvement epitomizes the depth and specificity mandated by these mechanisms. As we broaden our comprehension of the Golgi apparatus’s functions within the nervous system, we inch closer to deciphering its elaborate mechanisms and the repercussions of neuronal malfunctions. This enriched understanding not only propels foundational neuroscience research but also illuminates prospective therapeutic interventions against formidable neurodegenerative disorders, notably Alzheimer’s.

Exploring the myriad functions of the Golgi apparatus in neuronal health and pathology unveils transformative horizons in neuroscience. Grasping the intricate nexus between this pivotal organelle, neural circuitry, and the trajectory of neurological disorders could pave the way for innovative therapeutic strategies, especially in the context of Alzheimer’s, which afflicts countless individuals globally. A nuanced comprehension of the interplay between the GM130 protein and the Golgi apparatus might pioneer treatments with the potential to attenuate, halt, or even reverse disease progression.

Delving into the Golgi apparatus’s role in synaptic plasticity could wield profound implications for diverse cognitive dysfunctions and neuronal traumas, providing fresh insights into memory consolidation, learning, and neural rejuvenation.

Furthermore, elucidating the ramifications of Golgi apparatus fragmentation in viral pathologies could spotlight innovative therapeutic avenues for myriad infections. Beyond SARS-CoV-2, pathogens such as Zika, HIV, and influenza capitalize on the host cell’s Golgi apparatus. By intensifying research endeavors in these domains, potential retroviral therapeutics could emerge, offering significant impacts on formidable infections such as HIV or novel interventions against Zika. 

Prospective research trajectories ought to recognize this organelle’s integral role in cellular life cycles. Crafting novel pharmaceutical agents that modify the virus–Golgi interaction, whilst preserving cellular cytoarchitecture, could represent a groundbreaking scientific advancement.

As investigative endeavors into the Golgi’s myriad functionalities intensify, it becomes patently evident that this organelle transcends mere cellular infrastructure, asserting itself as a cornerstone in diverse biological and pathological paradigms. Progressing in our grasp of these dimensions demands a collective endeavor from the international scientific fraternity, paired with an unwavering dedication to unlocking the intricacies of this cellular nexus.

## Figures and Tables

**Figure 1 brainsci-13-01363-f001:**
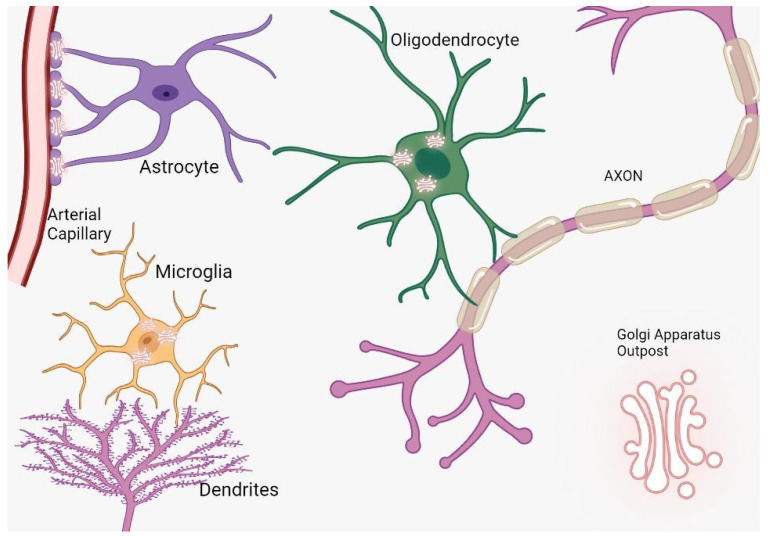
The distribution and function of Golgi outposts in different types of glial cells. In oligodendrocytes, these outposts are located in processes and the myelin sheath, where they nucleate new microtubules along radial and lamellar microtubules. In microglia, Golgi outposts contribute to the nucleation of new microtubules and the establishment of branched processes. In astrocytes, Golgi outposts are observed in endfeet contacting blood vessels as evidenced by electron microscopy, although their specific function is yet to be determined.

**Figure 2 brainsci-13-01363-f002:**
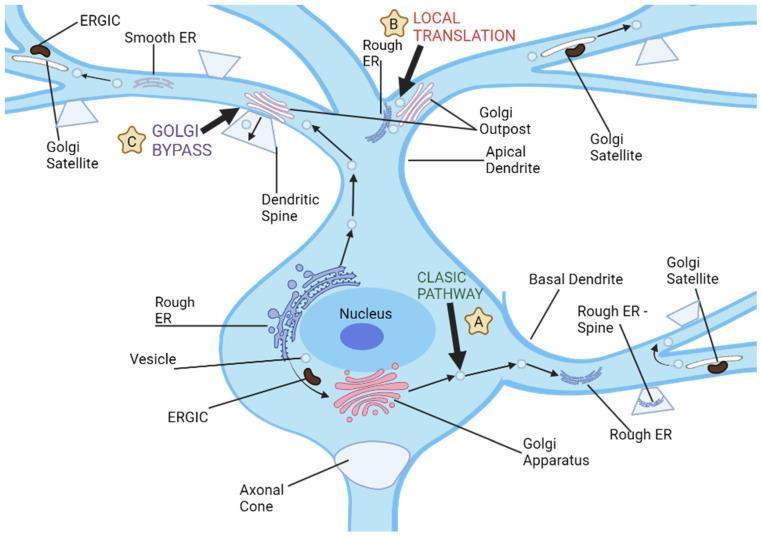
Neuronal dendrites provide multiple pathways for cargo transport and secretory processes, and there are three distinct routes used for dendritic cargo transportation. (**A**) The classic secretion pathway involves protein translation in neuronal cell bodies, followed by exit from endoplasmic reticulum and further processing by the ER-Golgi intermediate compartments and cell body Golgi. Vesicles then transport these proteins to dendrites or other parts of a neuron for storage and transport. (**B**) Local dendritic translation involves synthesizing proteins on dendritic rough ER before sending them off for further modification by dendritic Golgi outposts. After leaving Golgi outposts, proteins are transported via post-Golgi vesicles which may either travel along dendrites or fuse with either dendritic plasma membranes or synaptic spines. (**C**) Golgi bypass pathway allows cargo synthesized on the cell body ER to bypass its Golgi organelle and be transported directly to dendrites for modification by Golgi outposts before being packaged into post-Golgi vesicles destined for synapses or plasma membrane.

**Figure 3 brainsci-13-01363-f003:**
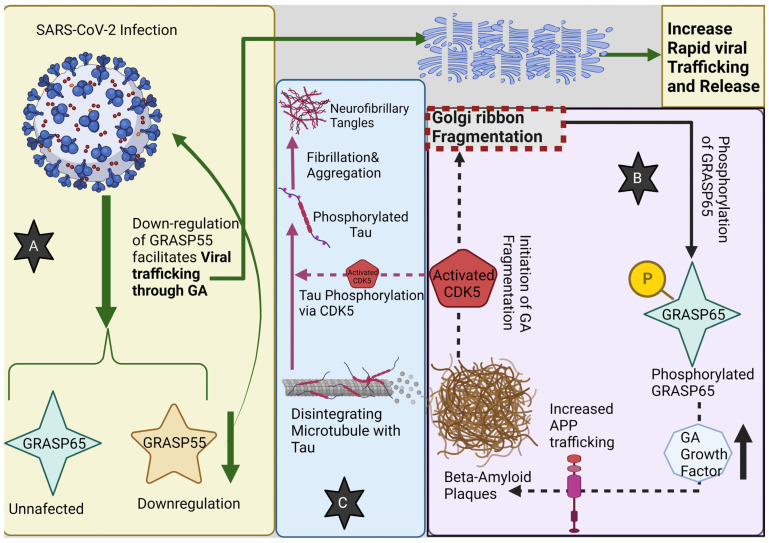
(**A**) SARS-CoV-2 infection down-regulates GRASP55, whereas GRASP65 remains unaffected. Down-regulation of GRASP55 facilitates viral trafficking through the Golgi apparatus (GA) leading to its fragmentation; GA fragmentation then leads to a rapid increase in viral trafficking and release. (**B**) Beta-amyloid plaques lead to GA fragmentation, through CDK5 activation, that in turn increases the phosphorylation of GRASP65, which will stimulate the GA growth factor synthesis and release, contributing to more APP (Amyloid Precursor Protein) trafficking, increasing the beta-amyloid plaque formation from Aβ oligomers. (**C**) Moreover, the activation of CDK5 contributes to Tau protein phosphorylation, thus promoting its fibrillation and aggregation into neurofibrillary tangles.

## Data Availability

All data are available via online libraries such as PubMed.

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
