# Peer review of "Unraveling the Multifaceted Role of the Golgi Apparatus: Insights into Neuronal Plasticity, Development, Neurogenesis, Alzheimer’s Disease, and SARS-CoV-2 Interactions"

_brainsci, 2023, doi:10.3390/brainsci13101363_

Round 1

Reviewer 1 Report

This review article briefly summarizes the role of the Golgi apparatus (GA) in neuronal development, plasticity, and Alzheimer disease (AD).  The authors also describe a hypothetical association between GA disruption induced by SARS-CoV-2 infection and AD. Below my comments and suggestions:

Major criticisms:

Redundancy, e.g., lines 38 vs 46, and 43 vs 55 … “protein sorting and modification”; … “regard to Golgi matrix protein GM130”, respectively. Lines 244 vs 245 … “migration”. Secretory pathway definition (lines 77, 152 and, 245). Definition (lines 77-79) is incorrect. In brief, the secretory pathway begins in the ER to produce newly synthesized proteins prior their passage through the GA for further processing, sorting, and export to later post Golgi/TGN compartments and final destinations such endosomes, lysosomes, and plasma membrane.  Line 435 vs 448 … “ribbon-like morphology

Reorganize and rename manuscript sections/headings-subheadings, e.g., in section 1. “Golgi Apparatus in Dendritic and Axonal Development” the authors just provide very basic information regarding the secretory pathway and the location of the GOPs in glial cells. GA involvement in dendritic formation and generation of GOPs in neural cells are described in section 2.

Minor:

Line 69; define Golgi apparatus outposts

Lines 143, 185…; Drosophila =>  Drosophila

Line 180; define mini-GAs, Golgi apparatus mini stacks?

Line 189; g-tubulin =>  gamma-tubulin

Line 201; what means Golgi apparatus cells?

Line 266; GOC => Conserved oligomeric Golgi complex (COG)

Line 409; a-mannosidase => alpha-mannosidase

Lines 490-91; sentence not understandable, rewrite.

Figure 3. Schematics represents fragmentation of Golgi stack cisternae not GA ribbon fragmentation/dispersion.

Already in comments for authors.

Author Response

We wish to express our gratitude for the time and effort dedicated to reviewing our manuscript titled "Unraveling the Multifaceted Role of the Golgi Apparatus: Insights into Neuronal Plasticity, Development, Neurogenesis, Alzheimer's Disease, and SARS-CoV-2 Interactions".

Redundancy of Information: We have scrutinized the entire manuscript and addressed all instances of redundant information to provide a more concise presentation.

Definition of the Secretory Pathway: We have rectified the definition, ensuring its accuracy and clarity in the context of the manuscript.

Reorganization and Renaming of Sections/Headings: Based on your recommendations, we have undertaken a comprehensive restructuring of our manuscript sections and revised the headings to better align with the content and enhance coherence for the reader.

In addition to addressing the major criticisms, we have meticulously attended to all minor criticisms to enhance the quality of the manuscript.

Reviewer 2 Report

The review article entitled “Unraveling the Multifaceted Role of the Golgi Apparatus: In-sights into Neuronal Plasticity, Development, Neurogenesis, Alzheimer's Disease, and SARS-CoV-2 Interactions” by  Toader et al  reviewed the  significance  of future research into the Golgi apparatus for potential therapeutic advances in  neurological disorders and viral diseases. Moreover, in the present review, the authors described the Golgi apparatus' prominent role in Alzheimer’s Disease, providing understanding into how its fragmentation could contribute to the dis

ease's pathophysiology. Moreover, the current review illustrated the newly emerging association between the Golgi apparatus and SARS-CoV-2 interactions, highlighting current findings that connect it to viral replication and the virus's lifecycle.

However, the author needs to address some of the important aspects of this review   

Such as

1. One of the limitations of the current review is the discussion about the recent investigation on neurological disorders relevant to COVID-19 is missing. Furthermore, if any findings are relevant the role of the Golgi apparatus in COVID-19 implications to neurological disorders needs to be discussed. However, authors have made reasonable attempts to show a newly emerging association between the Golgi apparatus and SARS-CoV-2 interactions, spotlighting recent findings that implicate it in viral replication and the virus's lifecycle.  

2. In the current review, authors discussed the GM 130 in sentence 43 then it was discussed in the 54th sentence.  For the viewer's attention and its flow, it should be addressed 44th sentence with relevant reference.

3. In the 233rd sentence relevant reference needs to be added.

3. At 244th sentence relevant reference needs to be included.  

4. In 253rd sentence authors were discussing the role of LIMK1 and PKD1  in TGN fission but the specific references were missing in this context.

5. In the 264th and 267th sentences reference pattern was different from the other references, please check it and correct it.

6. In the 277th sentence, the authors discuss Menkes disease, but the relevant reference was missing.

7. In the 316th  sentence authors are recommended to add the following reference to enhance the justification of the statement.

Mitochondrial Deficits Accompany Cognitive Decline Following Single Bilateral Intracerebroventricular Streptozotocin. Curr Alzheimer Res. 2015;12(8):785-95. doi: 10.2174/1567205012666150710112618 (PMID: 26159195)

8. At the 340th sentence relevant reference needs to be included.  

9. At 390 relevant references need to be added.

10. At 463 sentences please consider adding the following recent finding reference to enhance the justification of the statement.
Muscle-building supplement β-hydroxy β-methylbutyrate binds to PPARα to improve hippocampal functions in mice. Cell Rep. 2023 Jul 25;42(7):112717. doi: 10.1016/j.celrep.2023.112717. Epub 2023 Jul 11. PMID; 37437568

Minor editing of the English language required  

Author Response

Dear Reviewer,

We wish to express our gratitude for the time and effort dedicated to reviewing our manuscript titled "Unraveling the Multifaceted Role of the Golgi Apparatus: Insights into Neuronal Plasticity, Development, Neurogenesis, Alzheimer's Disease, and SARS-CoV-2 Interactions".

Reference Updates: We have thoroughly updated our references, ensuring their relevance and accuracy. In line with your suggestions, we have also incorporated recent studies pertinent to SARS-CoV-2 infections, enhancing the currency and depth of our review.

Reviewer 3 Report

1.       The abstract is quite long, which may make it less effective at quickly summarizing the main points of the article. It's generally recommended to keep abstracts concise and focused on the essential information.

2.       While the abstract covers various aspects of the Golgi apparatus's role in neurological contexts, it could benefit from clearer organization and a more explicit statement of the key findings or contributions of the article.

3.       The use of technical terms like "neurite outgrowth" and "SARS-CoV-2 interactions" without explanation might make the abstract less accessible to a broader audience. Consider providing brief explanations or definitions for such terms to enhance clarity.

4.       There is some repetition in the abstract, such as the repeated mention of the Golgi apparatus's role in different contexts. Streamlining the language and avoiding redundancy can make the abstract more concise and reader-friendly.

5.       The statement "implicate it in viral replication and the virus's lifecycle" is somewhat vague. It could be improved by specifying the exact mechanisms or processes involving the Golgi apparatus in SARS-CoV-2 interactions.

6.       The abstract could benefit from a more impactful concluding sentence that summarizes the overarching significance of the research and its potential implications.

7.       The introduction repeats some information, such as the Golgi apparatus's role in protein sorting and modification, in slightly different phrasing. This repetition can be condensed for a more concise and impactful introduction.

8.       While the introduction mentions "recent studies" and "recent evidence," it doesn't provide specific citations or references to these studies. Including citations would add credibility and allow readers to explore the source material for more in-depth information.

9.       The introduction could benefit from clearer organization. It starts by introducing the Golgi apparatus's role in neuronal development and synaptic plasticity, then transitions to its role in Alzheimer's disease. A clearer separation or transition between these topics might improve the flow.

10.    "Recent studies have shed new light on an unexpected link between the Golgi apparatus and Alzheimer's pathogenesis," are somewhat redundant and could be made more concise.

11.   The phrase "shaping axons and dendrites dynamics" could be clarified for better readability. Additionally, the introduction could provide brief explanations or definitions for technical terms like "Golgi matrix protein GM130" to ensure accessibility to a broader audience.

12.   The introduction could benefit from a concluding sentence that summarizes the main focus of the article and its significance in the field of neurobiology.

13.   The statements lack clear structure and organization. Information is presented in a somewhat fragmented manner, making it challenging to follow the logical flow of ideas.

14.   Citation and Attribution: While some statements reference specific studies or findings, others do not provide citations or references to support the claims made. Proper citation is crucial to establish the credibility of the information presented.

15.   Some concepts and ideas are repeated in different sections of the statements. This redundancy could be eliminated to make the text more concise.

16.   Several sentences are quite long and complex, which can hinder clarity. Breaking down complex ideas into simpler sentences and providing clear explanations can enhance comprehension.  

17.    The statements lack concluding sentences that summarize the main points or tie the information together. Adding concise summaries can help readers grasp the key takeaways.

18.   The statements transition between topics abruptly, making it challenging to follow a clear narrative. It's important to provide smoother transitions between different concepts.

minor editing

Author Response

Dear Reviewer,

We wish to express our gratitude for the time and effort dedicated to reviewing our manuscript titled "Unraveling the Multifaceted Role of the Golgi Apparatus: Insights into Neuronal Plasticity, Development, Neurogenesis, Alzheimer's Disease, and SARS-CoV-2 Interactions".

Abstract Revision: We have revised the abstract to encapsulate the essence of our review more effectively, keeping in mind your valuable suggestions.

Citations List: We have updated the citations list, ensuring comprehensive coverage and accuracy.

Introduction Criticism: Your feedback on the introduction was particularly constructive, and we have revamped it accordingly to set a clearer stage for our review.

Transition between Topics: The transitions between various topics have been rewritten to ensure a smoother narrative flow, enhancing the reader's journey through our review.

In conclusion, each of your comments has been instrumental in refining our manuscript, ensuring it achieves a higher standard of academic rigor and clarity. We deeply appreciate your invaluable feedback and the constructive criticisms provided.

We are hopeful that our revisions will make the manuscript worthy of publication, and once again, we extend our heartfelt gratitude for your significant contributions to enhancing the quality of our work.

Round 2

Reviewer 1 Report

The authors have reasonably addressed the issues raised in my previous review.

Reviewer 2 Report

Thank you for the authors to addressing my comments. There will be no further comments 

Minor editing of the English language is required